# AUDIO-AGENT: LEVERAGING LLMS FOR AUDIO GENERATION, EDITING AND COMPOSITION

## ABSTRACT

We introduce Audio-Agent, a multimodal framework for audio generation, editing and composition based on text or video inputs. Conventional approaches for text-to-audio (TTA) tasks often make single-pass inferences from text descriptions. While straightforward, this design struggles to produce high-quality audio when given complex text conditions. In our method, we utilize a pre-trained TTA diffusion network as the audio generation agent to work in tandem with GPT-4, which decomposes the text condition into atomic, specific instructions, and calls the agent for audio generation. Consequently, Audio-Agent generates high-quality audio that is closely aligned with the provided text or video while also supporting variable-length generation. For video-to-audio (VTA) tasks, most existing methods require training a timestamp detector to synchronize video events with generated audio, a process that can be tedious and time-consuming. We propose a simpler approach by fine-tuning a pre-trained Large Language Model (LLM), e.g., Gemma2-2B-it, to obtain both semantic and temporal conditions to bridge video and audio modality. Thus our framework provides a comprehensive solution for both TTA and VTA tasks without substantial computational overhead in training.

## 1 INTRODUCTION

Multimodal deep generative models have gained increasing attention these years. Essentially, the models are trained to perform tasks based on different kinds of input called modalities, mimicking how humans make decisions from different kinds of senses such as vision and smell Suzuki & Matsuo (2022). Compared to other generation tasks such as image generation or contextual understanding, audio generation is less intuitive as it is harder to precisely measure the generated sound quality using human ears. Additionally, previous works mainly focus on generating music-related audio, which is more structured compared to naturally occurring audio Copet et al. (2024); Melechovsky et al. (2023). Some recent works have focused on generating visually guided open-domain audio clips Chen et al. (2020); Zhou et al. (2018).

Recent researches on audio generation are mainly focused on text-to-audio generation (TTA) and video-to-audio generation (VTA). For TTA task Xue et al. (2024); Kreuk et al. (2022), current datasets lack high-quality text-audio pairs. Existing datasets such as AudioCaps Kim et al. (2019) or Clotho Drossos et al. (2020) usually contain multiple event descriptions mixed into one single sentence without fine-grained details and object bindings. This complicates training, particularly when handling long continuous signals with complex text conditions Huang et al. (2023). We define complex text conditions as long event descriptions containing a series of events without explicitly describing the sound, such as "A man enters his car and drives away". While previously not fully studied, this type of condition is more realistic as it does not require any detailed specification in terms of the characteristics of the audio result, offering more flexibility to the user and producer for areas such as movie dubbing and musical composition. If we train these models from scratch, it often demands extensive computational resources Liu et al. (2024); Ghosal et al. (2023).

The VTA task, or conditional Foley generation, remains unexplored until recently Wang et al. (2024); Zhang et al. (2024b). One main challenge is that video clips typically contain excessive visual information not always relevant to audio generation. Moreover, synchronization is hard between video and audio output, with recent solutions such as temporal masking Xie et al. (2024) proving inadequate for complex scenarios. Due to efficiency considerations, current methods often

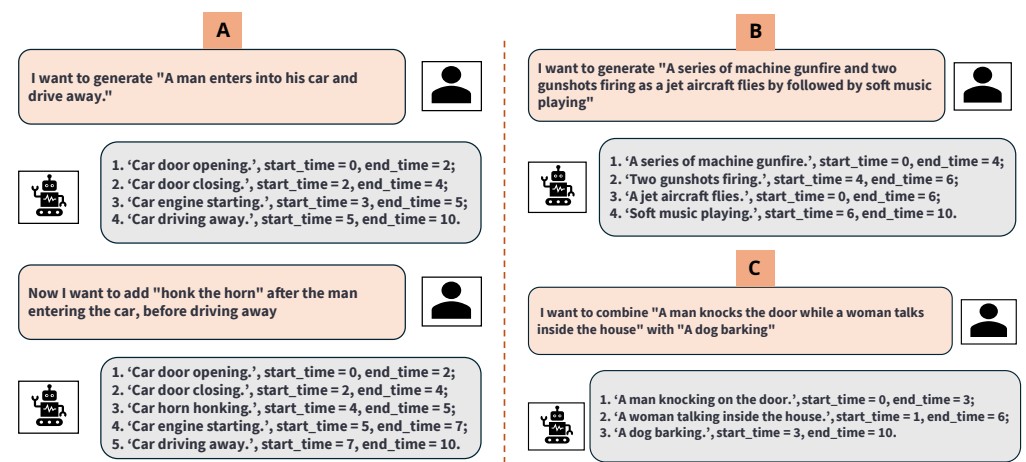

Figure 1: Example showing Audio-Agent's ability to generate, compose and edit multiple audio descriptions together. (A): Multi-turn editing; (B): Generation based on long description; (C): Multiple audio descriptions composition

encode video features by extracting a few random frames Xie et al. (2024); Dong et al. (2023), which hinders learning temporal information. Bridging the modality gap Liang et al. (2022) between video and audio thus becomes the key to solving the problem.

While achieving state-of-the-art results, conventional approaches often perform inference in a single pass based on a given text description. This approach struggles to produce high-quality audio when faced with complex or lengthy text conditions. In this paper, we introduce Audio-Agent, which breaks down intricate user inputs using GPT-4 into multiple generation steps. Each step includes a description along with start and end times to effectively guide the audio generation process. Our framework integrates two key tasks: Text-to-Audio (TTA) and Video-to-Audio (VTA). We leverage a pre-trained TTA diffusion model, Auffusion Xue et al. (2024), with essential adaptations, serving as the backbone for our generation process. In the TTA task, Auffusion focuses solely on generating simple, atomic text inputs. Our framework supports audio generation, editing, and composition, as illustrated in Figure 1. For the VTA task, we recognize that models such as GPT-4 and other large language models lack sufficient temporal understanding of video clips. To address this problem, we employ moderate fine-tuning to align the two modalities. We utilize the smaller Gemma2-2B-it model, which has 2 billion parameters, and fine-tune an adapter and a projection layer to convert visual inputs into semantic tokens. We then implement cross-attention guidance between the diffusion layers of Auffusion. This approach eliminates the need for additional training on a temporal detector, as the semantic tokens inherently contain time-aligned information.

The summary of our contributions is as follows: 1) we propose Audio-Agent which utilizes a pre-trained diffusion model as a generation agent, for both TTA and VTA tasks; 2) For TTA, Audio-Agent can handle complex text input, which is broken down into simple and atomic generation conditions for the diffusion model to make inference on; 3) For VTA, we fine-tune an open-source LLM (Gemma2-2B-it) to bridge the modality gap between video and audio modalities to align the underlying semantic and temporal information. Through extensive evaluation, our work demonstrates on-par results compared to the state-of-the-art task-specific models trained from scratch, while capable of producing high-quality audio given long and complex textual input. We hope our work can motivate more relevant works on multi-event long-condition TTA generation, which to our knowledge has not yet been fully explored despite its high potential in various content generations where high-quality audio is essential.

## 2 RELATED WORK

**LLM-based Agent Method** Recent progress in large language models has enabled relevant research on making LLM a brain or controller for the agent on performing various tasks, such as robot task planning and execution Driess et al. (2023) or software development Rawles et al. (2024); Yang et al. (2023). LLM demonstrates the capacity of zero-shot or few-shot generalization, making task transfer

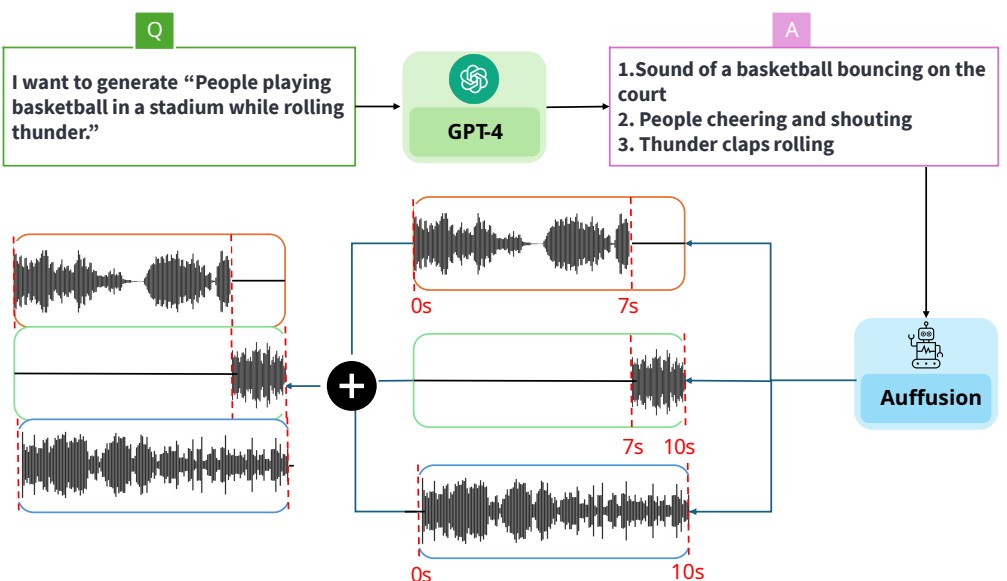

Figure 2: Overview of the TTA part. We use GPT-4 to convert a complex audio generation process into multiple generation steps and combine inference results.

possible without significant change of its parameters Xi et al. (2023). In our work, we harness the action-planning ability of LLM. Upon receiving the text condition from the user, LLM generates a plan with detailed steps on how to call the diffusion model which serves as a generation agent. By dividing the task into simpler sub-tasks, we can ensure the generation quality with fine-grained event control for TTA generation.

**Diffusion-based Audio Generation** AudioLDM Liu et al. (2024) is among the pioneering works that introduce the latent diffusion method to audio generation. Subsequent works such as Tango Ghosal et al. (2023) and Auffusion Xue et al. (2024) use pre-trained LLM such as Flan-T5 for text encoding, which has been widely adopted. We notice that this method can be seamlessly adapted to VTA tasks when we can find a similarly effective way of utilizing LLM for encoding the visual content. For the TTA task, we choose Auffusion as our generation agent due to its outstanding performance on fine-grained alignment between text and audio.

**Coarse-to-fine Audio Generation** Current works such as AudioLM Borsos et al. (2023), VALL-E Wang et al. (2023) and MusicLM Agostinelli et al. (2023) use multiple codebooks and Residual Vector Quantization (RVQ) Défossez et al. (2022) to create diverse audio representations. In AudioLM, the model first predicts semantic tokens that capture crucial information for overall audio quality, such as rhythm and intonation, while subsequent layers add details to enhance the richness of the generated sound. However, these discrete designs suffer from generation quality compared to their continuous-valued counterparts. Moreover, the model has to perform prediction over multi-layers, which inevitably increases computational demands for both training and inference Meng et al. (2024). In our case for the VTA task, we fine-tune an LLM to predict an intermediate discrete representation as semantic tokens using a language modeling approach. The discrete semantic tokens then serve as a condition for the diffusion model to generate continuous predictions. In this way, our method simplifies the generation procedure while maintaining the advantages of audio generation using the language modeling approach.

## 3 METHOD

Audio-Agent comprises three major components: 1) GPT-4 as a brain for action planning; 2) a lightweight LLM to convert video modality into semantic tokens; and 3) a pre-trained TTA diffusion model as the generation backbone. Our model structure is illustrated in Figure 2 and Figure 3.

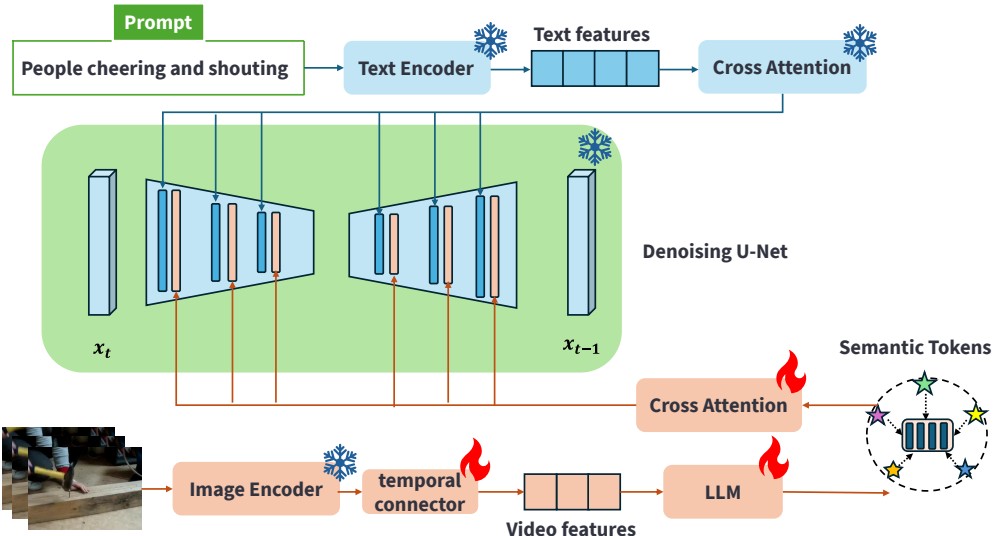

Figure 3: Overview of the generation backbone. We build on top of the pre-trained Auffusion model for both TTA and VTA generation.

## 3.1 PRELIMINARIES

**Audio Latent Diffusion Model** Recent research adapted the successful latent diffusion models from the image domain to the audio domain. A typical audio latent diffusion model such as Auffussion first converts the audio wave into mel spectrogram, followed by VAE encoding into the relevant latent space. Inference is the reverse process, where the predicted latent is decoded by VAE and then converted back from mel spectrogram into audio wave through a vocoder such as HiFi-GAN Kong et al. (2020). The latent diffusion process can be regarded as the same as the standard latent diffusion model on image generation Rombach et al. (2022).

**Semantic token** AudioLM Borsos et al. (2023) was among the first to propose a two-stage method for speech synthesis. In their method, the semantic tokens are derived from representations produced by an intermediate layer of w2v-BERT Chung et al. (2021). We choose an open-sourced HuBERT Hsu et al. (2021) model to produce the semantic representation, since HuBERT can model long-term temporal structure in a generative framework. Although only the smallest Hubert model has its quantizer released and open-sourced, we found that the released small model is already enough to assist the diffusion model in generating high-quality and temporally aligned predictions.

## 3.2 GPT-4 AS AN ACTION PLANNER FOR TTA TASK

Given a long, complex text condition, we ask GPT-4 to decompose the description into simple and atomic generation steps. GPT-4 has the freedom to decide how many steps to generate. We additionally restrict GPT-4 to keep the minimum number of necessary generation steps. This step instruction produces a good balance avoiding either extreme of being too abstract or too specific with unnecessary details. We also inform GPT-4 that the user may revise the text requirement in subsequent conversations so that our framework can perform multi-turn conversational generation. The output of GPT-4 consists of a JSON file, which contains a series of function calls of the agent model with text description provided. In addition, to support variable length generation and multi-event generation, GPT-4 also provides the start time and end time for each call which can overlap with each other. After obtaining the generation result for each step, we add waveforms together based on their time range. See Appendix A.1 for a prompt example.

## 3.3 AUDIO TOKENIZER AND VIDEO TOKENIZER

Following Kharitonov et al. (2021), we utilize the 9th layer of the Hubert-Base model to derive the semantic tokens. The quantizer of Hubert-Base contains 500 centroids. Given an audio clip

Table 1: Comparison of functionalities between recent audio generation framework. For AudioLDM2 and Auffusion half check marks are assigned because the corresponding model was trained only on 10 seconds of audio clips. In theory, it also supports long audio generation, but the quality is not assured, see Figure 5

| Method | VTA generation | TTA generation | | |
| | | Multi-turn editing | Composition | Long complex generation |
|---|---|---|---|---|
| Diff-Foley | ✓ | ✗ | ✗ | ✗ |
| FoleyCrafter | ✓ | ✗ | ✗ | ✗ |
| AudioLDM2 | ✗ | ✗ | ✗ | ✓̸ |
| Auffusion | ✗ | ✗ | ✗ | ✓̸ |
| Ours | ✓ | ✓ | ✓ | ✓ |

as ground truth, Hubert acts as an audio tokenizer that applies K-mean clustering and converts the audio into discrete semantic tokens, where each token has a value ranging from 0 to 499 to represent the respective centroids. Hubert-Base has a frame rate of 50Hz, thus a 10-second audio will result in 500 semantic tokens.

To efficiently capture both visual and temporal information while compressing the video data, we employ CLIP as a frame-wise feature tokenizer. CLIP is compatible with arbitrary frame sampling strategies, enabling a more flexible frame-to-video feature aggregation scheme as noted by Cheng et al. (2024). We pool the information within each frame to reduce the sequence size, resulting in a vector $f^r$ of size $N \times D$, where $N$ is the number of frames and $D$ is the CLIP hidden size. We set the frame rate to 21.5 Hz and use CLIP ViT-L/14 by default.

Inter-frame information is crucial for the model to achieve temporal alignment. Previous methods Iashin & Rahtu (2021); Du et al. (2023) require extracting both RGB and optical flow information within and across frames. In our design, we add a temporal connector after obtaining frame-wise features. The temporal connector consists of a 1D convolution block and a projection layer. The convolution block aggregates the inter-frame features together while preserving the temporal order. The projection layer projects the features into LLM's embedding space.

### 3.4 LLM FOR SEMANTIC TOKEN GENERATION ON VTA TASK

Semantic tokens allow us to represent continuous audio information in discrete semantic form. We denote the continuous audio ground truth as $a \in \mathbb{R}^{C \times L}$, where $C$ is the number of channels and $L$ is the time of the audio clip times sample rate. The Hubert audio tokenizer applies the K-means algorithm to convert the representation into LLM-aware acoustic tokens. Specifically, we obtain the indices $s \in \{0, ..., 499\}^N$ from the audio by comparing it with the encoded audio with centroids, and $N$ is the sequence length.

During training and inference, we feed the model with encoded video embedding and caption, together with the instruction prompt. To better differentiate the video input with text condition and instruction, we wrap the encoded video feature with special tokens as modality indicators. Specifically, we wrap the video caption with ⟨*Caption*⟩, ⟨*/Caption*⟩ indicators and video embedding in an embedded sequence of ⟨*Video*⟩, ⟨*/Video*⟩ indicators. In doing so, we avoid the possibility of confusing the LLM with different kinds of information. See Appendix A.2.

To jointly model different modalities in a unified model, we further extended the LLM's text vocabulary $V_t = \{v_i\}_{i=1}^{N_t}$ with acoustic vocabulary $V_a = \{v_j\}_{j=1}^{N_a}$. The acoustic vocabulary includes the modality indicators and a series of semantic tokens in the form of ⟨*AUD_X*⟩, where $X$ ranges from 0 to 499, the same as the number of centroids of the audio tokenizer. The extended audio-text vocabulary now becomes $V = \{V_t, V_a\}$.

To further elaborate on the conditional generation tasks performed by LLM: for the VTA task, the source input $X_v = \{x_e^i\}_{i=1}^N$ is a sequence of embeddings and $x_e \in \mathbb{R}^D$, where $D$ is the embedding dimension of LLM. Our LLM backbone is a decoder-only structure with the next token prediction method. The distribution of the predicted token in the first layer is given by $p_{\theta_{LLM}}(\mathbf{C}_1|X) =$

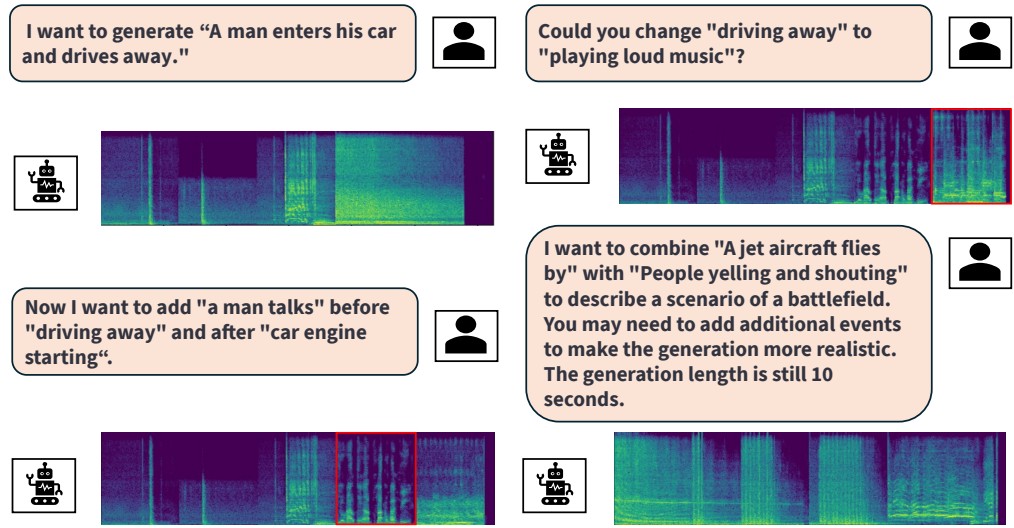

Figure 4: A demo example showing Audio-Agent's conversation ability: First turn: Audio Generation; second turn: Audio Insertion; third Turn: Audio Editing; last turn: Audio Composition with high-level semantic instructions. Audio-Agent can choose to respond based on previous turns or make independent generations.

$\prod_i p_{\theta_{LLM}}(c_1^i|X, \mathbf{C}_1^{<i})$ autoregressively. The objective has thus become:

$$\mathcal{L}_{LLM} = -\sum_{i=1}^{T'} \log p_{\theta_{LLM}}(c_1^i|X, \mathbf{C}_1^{<i}), \qquad (1)$$

where $T'$ is the number of semantic tokens generated by LLM, $\theta_{LLM}$ is the parameter of LLM, $c_1^i$ is the token generated at step $i$, $\mathbf{C}_1^{<i}$ are previous tokens, and $X$ is the input condition.

During inference, the LLM will autoregressively predict the next token until $\langle eos \rangle$ is generated. Our LLM thus serves as the bridge for connecting between modalities.

In our experiments, we use Gemma2-2B-it Team et al. (2024), a lightweight open-source LLM developed by Google, which is claimed to have comparable performance to a much larger variant Gemma-2-9B. We use Low-Rank Adaptor (LoRA) Hu et al. (2021) to finetune Gemma to make it understand vision/text conditions and generate audio tokens.

### 3.5 CONDITIONAL AUDIO GENERATION

The audio generation module contains a diffusion model, text-based cross-attention layers and visual-based cross-attention layers. See Figure 3. Given a query feature $Z$, text features $c_{txt}$ and visual features $c_{vis}$ the output for combining two types of cross-attention is defined as follows:

$$\mathbf{Z}^{new} = \text{Softmax}(\frac{\mathbf{Q}\mathbf{K}_{txt}^\top}{\sqrt{d}})\mathbf{V}_{txt} + \text{Softmax}(\frac{\mathbf{Q}(\mathbf{K}_{vis})^\top}{\sqrt{d}})\mathbf{V}_{vis}$$
$$\text{where } \mathbf{Q} = \mathbf{Z}\mathbf{W}_{txt}^q, \mathbf{K}_{txt} = c_{txt}\mathbf{W}_{txt}^k, \mathbf{V}_{txt} = c_{txt}\mathbf{W}_{txt}^v, \qquad (2)$$
$$\mathbf{K}_{vis} = c_{vis}\mathbf{W}_{vis}^k, \mathbf{V}_{vis} = c_{vis}\mathbf{W}_{vis}^v$$

The diffusion model and text-based cross-attention layers are from the pre-trained Auffusion model. During training, we keep the pre-trained part frozen. For the TTA task, we directly feed the step instructions as text conditions and arrange the output based on the start time and end time, as illustrated in Section 3.2. For the VTA task, after obtaining the semantic tokens, we fetch the centroids from the Hubert model according to the value indices as visual features. Similar to the text-based condition mechanism, we apply cross-attention on layers of the diffusion model. During inference, we introduce another parameter for controlling text and visual guidance:

$$\mathbf{Z}^{new} = \text{Attention}(\mathbf{Q}, \mathbf{K}_{txt}, \mathbf{V}_{txt}) + \lambda \cdot \text{Attention}(\mathbf{Q}, \mathbf{K}_{vis}, \mathbf{V}_{vis}) \qquad (3)$$

Thus the final objective for the diffusion process, which is similar to latent diffusion models, is

$$L_{\text{simple}} = \mathbb{E}_{\boldsymbol{x}_0, \boldsymbol{\epsilon}, \boldsymbol{c}_{txt}, \boldsymbol{c}_{vis}, t} \| \boldsymbol{\epsilon} - \boldsymbol{\epsilon}_\theta(\boldsymbol{x}_t, \boldsymbol{c}_{txt}, \boldsymbol{c}_{vis}, t) \|^2. \qquad (4)$$

Compared to IP-Adapter Ye et al. (2023), our method introduces the video modality into audio generation. Furthermore, since the semantic tokens already incorporate temporal information of the video, we do not need to train an extra timestamp detection module as done by FoleyCrafter Zhang et al. (2024b) to achieve temporal alignment.

### 3.6 IMPLEMENTATION DETAILS

For fine-tuning Gemma-2B-it, we set LoRA rank and alpha to be 64 with dropout to be 0.05. We separately train and fine-tune Gemma-2B-it, the projection layers and the cross-attention layers on the AVSync15 Zhang et al. (2024a) datasets. The training and evaluation are conducted on NVIDIA GeForce RTX 4090. Following Ye et al. (2023), we set the $\lambda$ to be 0.5 as default.

## 4 EXPERIMENTS

### 4.1 TRAINING DATASETS

For the TTA task, we evaluate our complex generation ability on AudioCaps Kim et al. (2019) dataset. We randomly choose either one caption from the test set or concatenate two of them together with the clause "followed by". To better compare with other models, we limit our generation length to the standard 10 seconds. Following Xue et al. (2024), we randomly selected 20 captions from each category for the generation. Additionally, to demonstrate Audio-Agent's ability to make inferences based on complex text conditions, we ask GPT to generate additional long event descriptions containing a series of events without explicitly describing the sound, such as "A man enters his car and drives away". The number of complex captions is also 20. The baseline methods include AudioGen-v2-medium Kreuk et al. (2022), AudioLDM2-large Liu et al. (2024) and Auffusion Xue et al. (2024).

We use AVSync15 for VTA task. AVSync15 is a curated dataset from VGGSound Sync Chen et al. (2021) that has 1500 high video-audio alignment pairs, which is ideal for training and demonstrating temporal alignment between video and audio. Same experiment setting as Zhang et al. (2024b) is used. To better facilitate evaluation, we include some audio generation results in the supplementary material.

### 4.2 EVALUATION METRICS

The evaluation metrics are summarized as follows: For the VTA task, we use the Frechet audio distance (FAD) to evaluate audio fidelity. Additionally, we utilize the MKL metric Iashin & Rahtu (2021) and CLIP similarity Wu et al. (2022) for audio-video relevance. Furthermore, to evaluate the synchronization of the generated audio in the video-to-audio setting, we use the same evaluation metrics as CondFoleyGen Du et al. (2023), namely # Onset Accuracy, and Onset AP. For the TTA task, we use CLAP similarity Wu et al. (2023)

### 4.3 EVALUATION AND COMPARISON

Audio-Agent outperforms other baseline methods on all TTA experiment settings, see Table 2. Additionally, our method outperforms the original Auffusion model by a significantly increasing margin as the text condition becomes longer and more complex. Specifically, we notice that with a longer text condition, AudioGen Kreuk et al. (2022), AudioLDM2 Liu et al. (2024) and Auffusion Xue et al. (2024) all exhibit missing out events. For example, if the text condition is multi-event such as "Pigeons cooing and bird wings flapping as footsteps shuffle on paper followed by motor sounds with male speaking", all the baseline methods fail to generate the motor sound at the

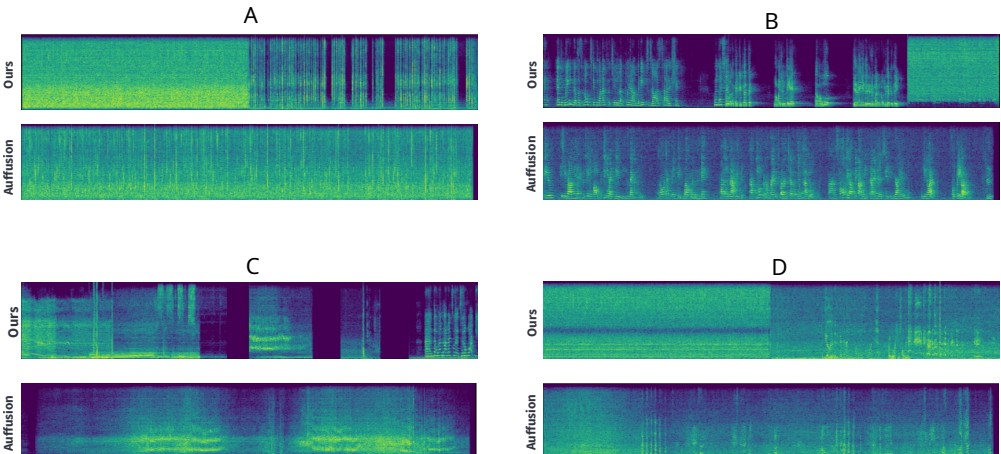

Figure 5: Comparison with baseline for TTA task. To demonstrate audio generation based on long complex text conditions, we ask the model to generate audio clips for 20 seconds. The text condition is drawn from the Two Captions category of Table 2: (A) A river stream of water flowing followed by typing on a computer keyboard; (B) A woman delivering a speech followed by a male speech and statics; (C) A vehicle engine revving then accelerating at a high rate as a metal surface is whipped followed by tires skidding followed by a door shutting and a female speaking; (D). Continuous white noise followed by a vehicle driving as a man and woman are talking and laughing; We can see that our method successfully generates multi-event audio at different times based on descriptions, while Auffusion mixes the generated audio.

end of the audio clip during evaluation. However, our method avoids this problem by utilizing GPT-4 as a brain/coordinator for caption analysis and generation planning, offering more fine-grained distinctions between events.

We also notice a significant drop for all methods on complex captions, since none of these methods has been trained on this type of text condition. Still, we find this type of text condition more practical in the real world, since it does not require explicit descriptions of the characteristics of sound, but rather describes the scenario for sound generation, offering more flexibility for the sound producer. We attach some examples of complex results that we used for evaluation in Appendix A.3.

For the VTA task, our method achieves better visual-audio synchronization compared to other baseline methods, while subpar the current state-of-the-art method in terms of generation audio quality, presented in Tables 3 and 4. We consider this reasonable as most of the other baseline methods have been trained on multiple larger datasets.

Specifically, we find that the temporal connector may negatively affect the generated audio quality on a small scale. However, for the evaluation of synchronization, we noticed a significant improvement after the temporal connector was applied, especially for the Onset AP. Without explicit training of a timestamp detector, our method achieves a better performance in terms of onset Acc and Onset AP, see Figure 6 for illustration.

Table 2: Evaluation for all baseline models on the TTA task, categorized by the type of text conditions.

| Method | Single Caption CLAP↑ | Two Captions CLAP↑ | Complex Captions CLAP↑ |
|---|---|---|---|
| AudioGen Kreuk et al. (2022) | 49.34% | 44.76% | 23.98% |
| AudioLDM2 Liu et al. (2024) | 47.04% | 36.03% | 23.33% |
| Auffusion Xue et al. (2024) | 50.91% | 45.90% | 14.40% |
| Ours | **55.17%** | **53.02%** | **24.06%** |

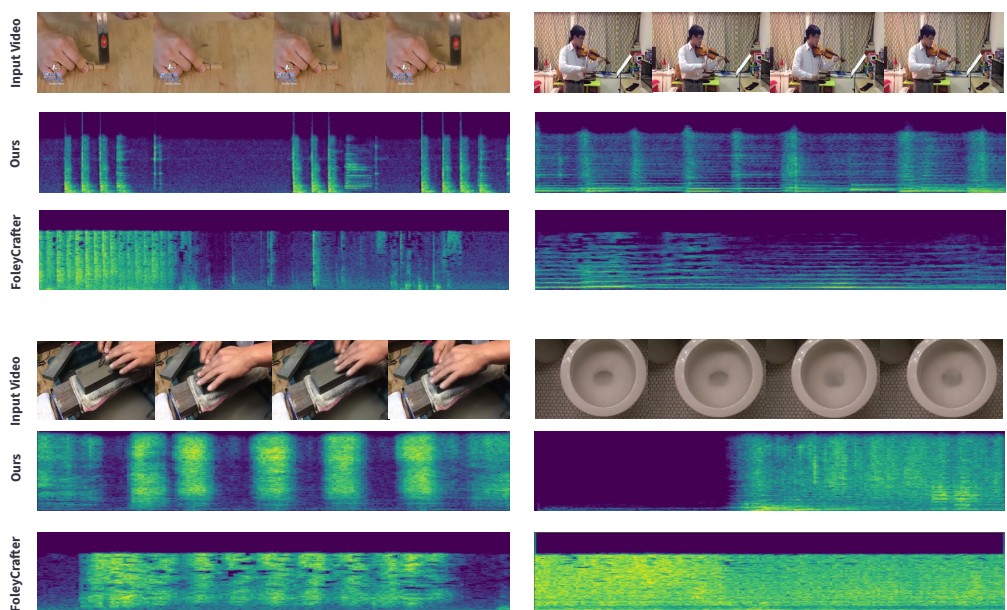

Figure 6: Comparison with baseline for VTA generation task. Compared to the baseline, the event occurrence is more explicit. Our method can produce audio that is more aligned and better synchronized with the input video.

Table 3: Quantitative evaluation on semantic alignment and audio quality. Specifically, Audio-Agent achieves on par performance versus state-of-the-art models in terms of Mean KL Divergence (MKL) Iashin & Rahtu (2021), CLIP Wu et al. (2022) and FID Heusel et al. (2017) on AVSync15 Zhang et al. (2024a).

| Method | MKL ↓ | CLIP ↑ | FID ↓ |
|---|---|---|---|
| SpecVQGAN (Inception) Iashin & Rahtu (2021) | 5.339 | 6.610 | 114.44 |
| SpecVQGAN (ResNet) Iashin & Rahtu (2021) | 3.603 | 6.474 | 75.56 |
| Diff-Foley Luo et al. (2024) | 1.963 | 10.38 | 65.77 |
| Seeing and Hearing Xing et al. (2024) | 2.547 | 2.033 | 65.82 |
| FoleyCrafter Zhang et al. (2024b) | **1.497** | **11.94** | **36.80** |
| Ours (without temporal connector) | 2.516 | 9.06 | 55.59 |
| Ours (with temporal connector) | 2.623 | 8.55 | 52.93 |

Table 4: Quantitative evaluation in terms of temporal synchronization. We report onset detection accuracy (Onset ACC) and average precision (Onset AP) for the generated audios on AVSync Zhang et al. (2024a), which provides onset timestamp labels for assessment, following previous studies Luo et al. (2024); Xie et al. (2024).

| Method | Onset ACC ↑ | Onset AP ↑ |
|---|---|---|
| SpecVQGAN(Inception) Iashin & Rahtu (2021) | 16.81 | 64.64 |
| SpecVQGAN(ResNet) Iashin & Rahtu (2021) | 26.74 | 63.18 |
| Diff-Foley Luo et al. (2024) | 21.18 | 66.55 |
| Seeing and Hearing Xing et al. (2024) | 20.95 | 60.33 |
| FoleyCrafter Zhang et al. (2024b) | 28.48 | 68.14 |
| Ours (without temporal connector) | 28.45 | 64.72 |
| Ours (with temporal connector) | **29.01** | **69.38** |

Table 5: Ablation study on AVSync15 dataset with different LoRA rank for semantic alignment and audio quality. During experiments, we keep the value of alpha the same as the rank.

| Method | Trainable Parameters | MKL ↓ | CLIP ↑ | FID ↓ |
|---|---|---|---|---|
| Ours (R=16) | 78.31MM | 2.702 | 8.42 | 58.426 |
| Ours (R=32) | 99.08MM | **2.543** | 8.49 | 55.197 |
| Ours (R=64) | 140.61MM | 2.623 | **8.55** | **52.929** |

Table 6: Ablation study on AVSync15 dataset with different LoRA rank in terms of temporal synchronization. During experiments, we keep the value of alpha the same as the rank.

| Method | Trainable Parameters | Onset ACC ↑ | Onset AP ↑ |
|---|---|---|---|
| Ours (R=16) | 78.31M | **29.74** | **70.63** |
| Ours (R=32) | 99.08M | 27.49 | 70.57 |
| Ours (R=64) | 140.61M | 29.01 | 69.38 |

## 4.4 ABLATION STUDIES

We include our ablation study on different LoRA rank values during LLM fine-tuning, see Tables 5 and 6. We found that an increase in trainable parameters sometimes does not necessarily improve the result. Notwithstanding, for a fair comparison, we use the rank value of 60 across all metrics. Additionally during training, we found that the training of the cross-attention layer can converge within 20,000 steps. We notice that the loss curve is not a reliable indicator of the model's performance. The model can achieve a good performance even when the loss curve remains flat.

## 5 CONCLUSION AND DISCUSSION

### 5.1 LIMITATION AND FUTURE WORK

Our framework experiences a drop in performance when given complex text conditions for the TTA task, which is more severe in other baseline methods. We believe it is a worthwhile direction in the future for understanding long complex captions with improved fine-grained distinctions between multiple events. We may also utilize the LLM's versatility involving audio captioning tasks and video captioning tasks. The above are worthwhile future directions to explore.

### 5.2 CONCLUSION

In this paper, we present Audio-Agent, a multimodal framework for both text-to-audio and video-to-audio tasks. Our model offers a conversation-based method for audio generation, editing and composition, facilitating audio generation conditioned on multievent complex descriptions. For the video-to-audio task, we propose an efficient method to achieve visual synchronization. Through extensive experiments, we show that our model can synthesize high-fidelity audio, ensuring semantic alignment with input. Additionally, our work takes an initial, significant step toward multi-event long-condition TTA generation which has not been fully explored.

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

# A   APPENDIX

## A.1   PROMPT EXAMPLE FOR TTA TASK

We provide our prompt instruction in Table 7 and in context examples in Tables 8 and 9.

## A.2   PROMPT EXAMPLE FOR VTA TASK

We provide our prompt instruction in Table 10. The prompt format follows the requirement from Gemma2-2B-it.

## A.3   COMPLEX CAPTIONS FOR TTA TASK

We provide examples of GPT-generated complex captions in Table 11 that we use for TTA task evaluation.

Table 7: Our prompt instruction for TTA generation

**You are a dialog agent that assists users in generating audio through
    conversation. The user begins by describing the audio they envision,
    and you help translate this description into multiple audio captions
    suitable for generating. You have a powerful tool at your disposal,
    Auffusion, which can generate simple, atomic audio based on textual
    descriptions. Your task is to determine how best to utilize this
    tool, which may involve multiple calls to Auffusion to produce a
    complex audio sequence composed of simpler audio.**

**Here are 10 examples of the types of descriptions Auffusion was
    trained on. These should guide you in understanding what constitutes
    a    simple   and   atomic   motion:**
1. A muddled noise of broken channel of the TV.
2. A person is turning a map over and over.
3. Several barnyard animals mooing in a barn.
4. An office chair is squeaking.
5. A flying bee is buzzing loudly around an object.
6. Thunder claps far in the distance.
7. Something goes round that is playing its song.
8. A paper printer is printing off multiple pages.
9. A person is making noise by tapping their fingernails on a solid
    surface.
10.A person crunches through dried leaves on the ground.

**Instructions:**
1. **User-Provided Description**: The user's description will include
    both straightforward and complex descriptions of audio. The user may
    also provide multiple descriptions and ask you to combine them
    together.
2. **Auffusion Invocation**: For each audio description, you must decide
    how to break down the description into simple, atomic audio. Invoke
    the Auffusion API to generate each component of the audio sequence.
    Ensure that each call focuses on a straightforward, non-elaborate
    audio description.
3. **Plan Generation**: Your response should include a step-by-step plan
    detailing each call to Auffusion necessary to create the complete
    audio sequence.
4. **Requirement**:
4.1. You should include the start_time and end_time in this call. The
    audio length is 10 seconds, and thus you should have at least one
    call having end_time=10.
4.2. If the user input has multiple events or asks to combine multiple
    description together, you should have overlapping audios happening
    in the same range of time. There should have less than three audios
    in the same time. Overlapping means one audio having smaller
    start_time than another audio's end_time
4.3. You're free to generate as many as calls you like, but please keep
    the minimum number of calls.

**Response Format:**
- You should only respond in JSON format, following this template:
```json
{
  "plan": "A numbered list of steps to take that conveys the long-term
    plan"
}
```

Table 8: Our in-context examples for TTA generation.

```
**Examples:**

**Example 1:**
- **User Input**: I want to generate "A clap of thunder coupled with the
  running water".
- **Your Output**:
```json
{
  "plan": "1. Auffusion.generate('A clap of
      thunders.',start_time=2,end_time=5); 2. Auffusion.generate('Rain
      pouring outside.',start_time=0, end_time=10)"
}
```

**Example 2:**
- **User Input**: I want to combine "Buzzing and humming of a motor"
  with "A man speaking" together
- **Your Output**:
```json
{
  "plan": "1. Auffusion.generate('A motor buzzing and
      humming',start_time=0,end_time=10); 2. Auffusion.generate('A man
      speaking.',start_time=3,end_time=6)"
}
```

**Example 3:**
- **User Input**: I want to generate "A series of machine gunfire and
  two gunshots firing as a jet aircraft flies by followed by soft
  music playing"
- **Your Output**:
```json
{
  "plan": "1. Auffusion.generate('A series of machine
      gunfire.',start_time=0,end_time=4); 2. Auffusion.generate('Two
      gunshots firing.',start_time=4,end_time=6); 3.
      Auffusion.generate('A jet aircraft
      flies.',start_time=0,end_time=6); 4. Auffusion.generate('Soft
      music playing.',start_time=6,end_time=10)"
}
```
```

Table 9: Our in-context examples for TTA generation (continue).

```
**Example 4:**
- **User Input**: I want to generate "A crowd of people playing
    basketball game."
- **Your Output**:
```json
{
  "plan": "1. Auffusion.generate('Sound of a basketball bouncing on the
      court.',start_time=0, end_time=7); 2. Auffusion.generate('A ball
      hit the basket',start_time=5, end_time=7); 3.
      Auffusion.generate('People cheering and shouting.',start_time=7,
      end_time=10)"
}
```
- **Followed up User Input**: I want to change it to "people playing
    table tennis".
- **Your Output**:
```json
{
  "plan": "1. Auffusion.generate('Sound of a table tennis ball bouncing
      on the table.',start_time=0,end_time=7); 2.
      Auffusion.generate('People cheering and
      shouting.',start_time=7,end_time=10)"
}
```
```

Table 10: Our prompt instruction for VTA generation

```
<start_of_turn>user
You are an intelligent audio generator for videos.
You don t need to generate the videos themselves but need to generate
    the audio suitable for the video, with sementic coherence and
    temporal alignment.
I'll give you the video embedding enclosed by <Video></Video>, also the
    video caption enclosed by <Caption></Caption>.
Your goal is to generate the audio indices for the video
You only need to output audio indices, such as <AUD_x>, where x is the
    index number.

Your turn:
Given the video <Video><VideoHere></Video> and the video caption
    <Caption><CaptionHere></Caption>, the accompanied audio for the
    video is:

<end_of_turn>
<start_of_turn>model
```

Table 11: Examples of our complex caption for TTA generation

```
1. A man enters his car and drives away
2. A couple decorates a room, hangs pictures, and admires their work.
3. A mechanic inspects a car, changes the oil, and test drives the
    vehicle.
4. A group of kids play hide and seek in a large, old house.
5. A woman packs a suitcase, locks her house, and walks to the bus
    station.
```

