# OpenReview forum: "Audio-Agent: Leveraging LLMs For Audio Generation, Editing and Composition"
_ICLR.cc/2025/Conference — ICLR 2025 Conference Withdrawn Submission_

### Official Review · Reviewer_w4th · 2024-11-02

**Soundness:** 4
**Presentation:** 3
**Contribution:** 3
**Rating:** 8
**Confidence:** 4

**Summary:**

This paper introduces Audio-Agent, a framework designed for audio generation, editing, and composition based on text or video inputs. The system leverages GPT-4 to decompose complex text descriptions into simpler, atomic instructions, which are then processed by a pre-trained text-to-audio diffusion model to generate audio. For video-to-audio tasks, the authors fine-tune a smaller language model, such as Gemma2-2B-it, to bridge the semantic and temporal gaps between video and audio modalities without the need for additional timestamp detectors. This approach enables high-quality audio generation that closely aligns with the provided text or video input, supports variable-length generation, and offers a user-friendly, conversation-style interaction for audio editing and composition. The framework demonstrates competitive performance compared to state-of-the-art models in both text-to-audio and video-to-audio tasks while reducing computational overhead.

**Strengths:**

1. The paper effectively utilizes GPT-4 to handle complex text conditions by breaking them down into simpler generation steps. This significantly improves multi-event audio generation and the alignment between input conditions and generated audio.

2. The proposed method offers a comprehensive solution for both text-to-audio and video-to-audio generation without the need for substantial additional training or computational resources.

3. The framework supports conversation-style interaction, enabling users to generate, edit, and compose audio through multi-turn dialogues. This design enhances usability and accessibility.

4. By leveraging existing pre-trained models like GPT-4 and Gemma2-2B-it, the system reduces the need for extensive computational resources and training from scratch.

5. In video-to-audio tasks, fine-tuning a pre-trained language model to capture semantic and temporal information removes the necessity for a separate timestamp detection module, simplifying the model architecture.

**Weaknesses:**

1. The framework relies on GPT-4, which is not open-source and may have accessibility or licensing constraints. This dependence could limit reproducibility and wider adoption in the research community.

2. The framework's complexity might make it challenging to generalize or scale to longer tasks or more extensive datasets. Handling longer and more complicated tasks may encounter upper limits in the current design.

3. The involvement of large language models like GPT-4 in the multi-step decomposition process might introduce latency during inference, affecting suitability for real-time applications.

4. While the authors claim reduced computational overhead, the paper lacks detailed benchmarks or comparisons regarding parameters and inference time relative to other state-of-the-art models.

**Questions:**

1. Can GPT-4 be replaced with open-source language models to improve accessibility and reproducibility? What impact would this substitution have on the model's performance?

2. How efficient is Audio-Agent in terms of the number of parameters and inference time compared to other state-of-the-art models? Can the authors provide benchmarks or comparisons?

3. How does the framework handle longer and more complicated tasks? Is there an upper limit to the length or complexity of the input that Audio-Agent can process effectively?

4. Given the potential latency introduced by GPT-4, how suitable is Audio-Agent for real-time audio generation or editing tasks?

5. What is the impact on performance if a smaller or less powerful language model replaces GPT-4 for text decomposition? Is GPT-4 essential, or can similar results be achieved with more accessible models?

6. How does the framework handle user inputs that require fine-grained control over audio properties, such as pitch or tempo? Can users influence these aspects through the input text?

---

### Official Review · Reviewer_tnN9 · 2024-11-02

**Soundness:** 2
**Presentation:** 2
**Contribution:** 2
**Rating:** 3
**Confidence:** 4

**Summary:**

This paper proposed Audio-Agent, leveraging the power of pre-trained LLM in audio generation and related tasks.
Audio-Agent can deal with both text-to-audio generation (TTA) and video-to-audio generation (VTA).

**Strengths:**

1. The application of Audio-Agent is practical. The conversation example shown in Figure 4 is impressive.
2. The paper faithfully presents the result, good and bad

**Weaknesses:**

1. The idea of employing a pre-trained LLM (e.g. GPT-4) to convert a complex description into multiple simple and atomic sound events has already been proposed in Make-an-audio 2 [1], which significantly undermines the novelty of this proposed work.
2. The paper's explanation of semantic tokens is unclear and confusing. Various types of semantic tokens are mentioned, including visual semantic tokens (L88), audio semantic tokens (L192), and textual semantic tokens (L262-263), but the relationship between these and the "semantic tokens" depicted in Figure 3 is not clearly established, along with why such varied descriptions are necessary.
3. Simply combining waveforms during the audio generation process can lead to abrupt changes in volume, adversely affecting the transition smoothness and volume balance of the final audio output. Some cases in the supplementary materials further corroborate this issue.
4. The results presented in Table 3 appear to be sub-optimal, lacking a sufficient analytical explanation for the performance presented. Additionally, the audio quality of the video-to-audio generation examples in the supplementary materials is relatively poor compared to the baseline, which raises concerns about the efficacy of the proposed methods.

References:
[1] Huang, J., Ren, Y., Huang, R., Yang, D., Ye, Z., Zhang, C., ... & Zhao, Z. (2023). Make-an-audio 2: Temporal-enhanced text-to-audio generation. arXiv preprint arXiv:2305.18474.

**Questions:**

See the above weaknesses.

---

### Official Review · Reviewer_McwQ · 2024-11-04

**Soundness:** 3
**Presentation:** 3
**Contribution:** 2
**Rating:** 5
**Confidence:** 4

**Summary:**

This paper proposes an agent system, called Audio-Agent, for Text-To-Audio and Video-To-Audio tasks. Particularly it addresses challenges of generating high-quality audio from complex and lengthy text prompt. In TTA, the system leverages GPT4 to break down a complex text prompt into simple prompts so that the diffusion model can handle easily. For VTA, a fine-tuned LLM is introduced to align the semantic and temporal structure between video and audio modalities. The proposed agent system is evaluated and compared against other state-of-the-art methods.

**Strengths:**

In TTA tasks, the capabilities of GPT-4 are shown for translating complex text prompt into simple and actionable prompts. The experimental results show the effectiveness of this approach.
In VTA tasks, the paper provides the details of finetuning an LLM and temporal connectors. It is found that some of the metrics is improved over the state-of-the-art methods.

**Weaknesses:**

This paper is rated as marginally below acceptance, since there are a few unclear points which would impact on the paper correctness and quality. The arguments are listed in Questions below.

**Questions:**

1.	3.4 LLM for semantic token generation on VTA Task: It is said that video caption is fed to the model during training and inference. If it is true, a) Fig 3 should be updated with video caption input, b) Table3/4 should indicate what inputs are taken in all the methods for clarity, c) comments should be made about comparing the proposed system to other methods which do not have the caption input.
2.	Table 2: It is described that, in the training, two-caption is generated by concatenating two prompts with the clause “followed by”. How about test data for the two-caption case? If the same process (concatenate + “followed by”) is applied to the test data, would it become known-to-work with GPT-4 (always get correct splits), should many variations of prompt be tested?
3.	4.3 Evaluation and comparison: What test datasets are used in the evaluation? No information can be found.

Things to improve the paper that did not impact the score:
1.	It would be helpful for readers to share sample audio clips to listen. Do you have a plan?
2.	Some of readers will wonder how the system would work with combined scenarios such as VTA + complex and lengthy text prompt. Do you have any insight?

---

### Official Review · Reviewer_YFVH · 2024-11-05

**Soundness:** 2
**Presentation:** 2
**Contribution:** 1
**Rating:** 1
**Confidence:** 5

**Summary:**

This work introduces Audio-Agent, an LLM-based framework for audio content creation, comprising two main components. For text-to-audio (TTA) tasks, Audio-Agent leverages GPT-4 to decompose input text instructions into multiple potential audio candidates arranged in temporal order. These candidates are then iteratively generated using Affusion and concatenated to produce the final output. For visual-to-audio (VTA) tasks, a Gemma-based LLM is employed to generate HuBERT semantic tokens conditioned on CLIP visual features, which serve as input for a diffusion-based audio generative model. The TTA component operates in a training-free manner, while the VTA component is trained on the AVSync15 dataset. Evaluation on both TTA and VTA tasks demonstrates the effectiveness of the proposed approach.

**Strengths:**

1. A multimodal framework for audio generation, offering multiple functionalities across tasks.
2. Good results are achieved on text-to-audio (TTA) and text-to-visual (T2V) tasks, outperforming some baseline methods.

**Weaknesses:**

1. The key idea behind Audio-Agent is to use an LLM to decompose complex text instructions, allowing for the generation of intricate audio content with precise control. However, this concept is not entirely original, as the authors have not referenced prior work (e.g., WavJourney, WavCraft) that introduced similar approaches. In addition, WavJourney offers more interactive content creation with complex instructions, such as storytelling and spatial-temporal composition.

2. The approach also lacks novelty for the VTA task, as similar methods, such as FolyGen and V2meow, are already present in the literature. For similar results, One could integrate a state-of-the-art VTA model into existing frameworks like WavJourney or WavCraft.

3. The choice of HuBERT as semantic tokens is questionable, as it’s pre-trained on speech-heavy datasets (e.g., Libri-Light) and fine-tuned for ASR tasks. More suitable alternatives like SemantiCodec, Encodec, or BEATs, which capture general audio representations. However, they are neither discussed nor ablated by the authors.

References:

[1]: WavJourney: Compositional Audio Creation with Large Language Models (https://arxiv.org/pdf/2307.14335)

[2]: WavCraft: Audio Editing and generation with Large Language Models (https://arxiv.org/pdf/2403.09527)

**Questions:**

Tiny comments:

1. In Section 4.1 Training Dataset, the authors combine details of the training and test sets.

---

### Note · Authors · 2024-11-14

I have read and agree with the venue's withdrawal policy on behalf of myself and my co-authors.